# TV Interaction as a Non-Invasive Sensor for Monitoring Elderly Well-Being at Home

**DOI:** 10.3390/s21206897

**Published:** 2021-10-18

**Authors:** Jorge Abreu, Rita Oliveira, Angel Garcia-Crespo, Roxana Rodriguez-Goncalves

**Affiliations:** 1DigiMedia, Department of Communication and Art, Campus Universitário de Santiago, University of Aveiro, 3810-193 Aveiro, Portugal; ritaoliveira@ua.pt; 2Innovation Promotion and Technological Development Institute, University Carlos III of Madrid, 28911 Leganes, Spain; angel.garcia@uc3m.es (A.G.-C.); roxrodri@inf.uc3m.es (R.R.-G.)

**Keywords:** TV, monitoring, elderly, non-invasive sensor, well-being, home, hazard situations

## Abstract

The number of technical solutions to remotely monitoring elderly citizens and detecting hazard situations has been increasing in the last few years. These solutions have dual purposes: to provide a feeling of safety to the elderly and to inform their relatives about potential risky situations, such as falls, forgotten medication, and other unexpected deviations from daily routine. Most of these solutions are based on IoT (Internet of Things) and dedicated sensors that need to be installed at the elderly’s houses, hampering mass adoption. This justifies the search for non-invasive technical alternatives with smooth integration that relying only on existent devices, without the need for any additional installations. Therefore, this paper presents the SecurHome TV ecosystem, a technical solution based on the elderly’s interactions with their TV sets—one of the most used devices in their daily lives—acting as a non-invasive sensor enabling one to detect potential hazardous situations through an elaborated warning algorithm. Thus, this paper describes in detail the SecurHome TV ecosystem, with special emphasis on the warning algorithm, and reports on its validation process. We conclude that notwithstanding some constraints while setting the user’s pattern, either upon the cold start of the application or after an innocuous change in the user’s TV routine, the algorithm detects most hazardous situations contributing to monitor elderly well-being at home.

## 1. Introduction

The monitoring of elderly people, preferably in a discreet and non-invasive way, has the potential to automatically identify accidents (such as falls or sudden illnesses), or even more importantly, to alert caregivers of potential risk situations, allowing them to take appropriate action to avoid serious accidents (problems arising from a lack of medication, physical disorientation, robberies, etc.). The issue of monitoring relates to the concepts of tele-care, tele-assistance, and telemedicine, which involve aspects associated with support and healthcare for the elderly when they are at home. These skills are ensured by technological means that allow the transfer—between the community (in a broad sense) and the patient—of information regarding diagnostics, therapy, and monitoring services [1].

The growing importance of tele-care systems (they help people to remain independent in their own homes) is corroborated by several studies [2,3,4]. In fact, nowadays these systems are even more pertinent and an effective option to fight adversities related to the COVID-19 pandemic [5].

It is also important to mention that tele-care, which does not replace face-to-face care, is taking an integrative and complementary role to increase the quality of the healthcare services provided. Technologically, tele-care systems are normally supported by various devices and tailor-made solutions, among which vital signs, fall, and presence sensors, along with fixed and mobile terminals, stand out. Additionally, there is an interesting window of opportunity to include television, in its interactive form (supported by bidirectional infrastructures as common IPTV or CATV networks), in this list of devices. We could use the interactions with the TV to obtain relevant data without the need for additional sensors.

In this sense, remote monitoring and tele-care solutions that are based on television—a device already available in the vast majority of homes and widely used by seniors [6]—have advantages that could be crucial for the generalization of these systems. Moreover, this approach, if properly implemented, can have dual advantages. First, it can directly contribute to the intended purpose: to support the monitoring of elderly people at home in a discreet, non-invasive way, based on the monitoring of their daily activity, without requiring changes in the home ecosystem. Second, it may allow caregivers to be automatically alerted to situations of potential risk, translated by deviations from the daily pattern of watching and interacting with the television, due to problems caused by sudden illness, physical disorientation, burglary, or any other incident that modifies typical activity, allowing caregivers to take timely action to avoid serious accidents.

Most solutions focus on the use of specific sensors that need to be installed in the elderly’s homes. They require some capital and ongoing costs and are a disruptive and even invasive technological intervention from the perspective of elderly people. Thus, we thought it reasonable to find a solution that avoids the problems mentioned by being transparent with the user and not requiring the installation of any type of sensor. The SecurHome TV ecosystem is what resulted. The idea came from the fact that the elderly are remarkably heavy users of television, and this simple device has the potential to operate as an invisible well-being sensor, and the entire solution works on top of a commercial IPTV system.

The SecurHome TV ecosystem, supported by the iNeighbour TV system [7], aims to alert caregivers to potential risk situations in which elderly people find themselves, using their domestic activity as starting point. For this purpose, the ecosystem is composed of three main components: (i) an interactive TV application that monitors, in a non-intrusive way, the senior’s interactions with the TV; (ii) a Web application for the caregiver, which allows him/her to understand the context associated with alerts; and (iii) a Web administration portal that, in addition to managing users’ accounts, allows scheduling daily reminders for medication, appointments, and medical exams. Based on these components, the SecurHome TV ecosystem allows for the monitoring the senior’s domestic activity, through his normal TV set; and detecting deviations from his patterns of TV consumption and interactions with the system, and delays in taking medication, which are duly considered through an algorithm for issuing alerts.

In the next section of the paper, a review of the literature, which identifies systems that help monitor the domestic activity of the elderly and pinpoints a lack of non-intrusive approaches based on the interaction with the TV, is presented. In Section 3, the components of the SecurHome TV ecosystem are described. Then we present the algorithm used to deliver alerts to caregivers in Section 4. Additionally, in Section 4, the algorithm validation process and the SecurHome TV API, which complements the detention algorithm, are presented. Section 5 provides the discussion of the results obtained during algorithm validation, and Section 6 presents the conclusions and future work.

## 2. Literature Review

The elderly and people with special needs constitute the main part of the target audience of remote monitoring and tele-care systems, especially the most recent ones integrating IoT approaches through various devices and sensors [8]. We are witnessing in Portugal, as in many other countries, growing penetration of these systems in elderly households—for example, the tele-assistance service provided by the Portuguese Red Cross (PRC) to guarantee care for the elderly [9]. This system integrates a bracelet or necklace connected to a landline telephone terminal, with an alarm button that, when pressed by the user, anywhere in the house, establishes instant telephone contact with the PRC’s Contact Center. This system can also contain a set of sensors (fall, smoke, gas, and security zone delimiter). It is enhanced with the possibility of georeferencing the user, adding new monitoring capabilities. In addition to this integrated PRC solution, there are still others in use in Portugal that use similar technologies, such as Comfort Keepers [10], which include, in addition to a bracelet with fall detection and GPS, a communication module that sends notifications to a mobile application for the caregiver. Additionally, the company True-Kare [11] developed a solution that is divided into two components: a True-Kare mobile phone, used as an ordinary mobile phone, but with design adjustments that make it easier to be used by elderly; and a Web application that is managed by a caregiver to support the True-Kare mobile phone user. Through this portal, the caregiver can obtain information regarding the health of their dependent and schedule medical appointments and medication. The mobile phone can be connected with other sensors, such as a blood pressure monitor.

This type of approach always implies that one or more sensors are transported or manipulated by the elderly, which, in addition to implying the (sometimes expensive) acquisition of hardware, does not always represent an efficient and comfortable solution. On the other hand, there are studies of solutions that allow the monitoring of the elderly’s activity through sensors, but in a non-invasive way. Some examples of these studies are presented below.

Alcalá et al. [12] suggested a novel technique to monitor human activity, based on non-intrusive load monitoring, capable of complying with the elderly people’s demands. The system uses smart meter data, which involves a low level of intrusiveness and is a potentially massive deployment. The authors used mathematic models and theories in the system’s assessment, and the results are promising for implementing the system in real context [13]. Charlon et al. [14] proposed a multi-sensor monitoring system for the elderly with cognitive disabilities. The system consists of a motion sensor network deployed in different home areas and in an electronic patch worn by elderly to detect falls. The system also benefits from an algorithm that detects abnormal situations and alerts the caregiver in real time through a Web application that displays the elderly person’s movements and alarms. Oliver et al. [15] proposed an ambient intelligence environment for cognitive rehabilitation at home, combining physical and cognitive activities. The authors implemented a system with smart sensors and actuators that is complemented with a remote monitoring tool, so that the therapist can supervise the patient’s exercises. Marcelino et al. [16] implemented an ecosystem to promote the physical, emotional, and psychic health and well-being of the elderly. The solution monitors vital signs, but also environmental parameters and behavior patterns, trying to find danger situations and predict harmful issues. The system triggers various alert levels, and the caregivers are notified according to the correspondent risks. Arshad et al. [17] proposed a system that uses a capacitive based sensor to monitor the daily activities of elderly people living alone to prevent and detect falls. The system is based on the level of disturbance in the electromagnetic field when a person approaches the sensor, which is embedded on the floor’s surface, making the system non-intrusive. Zhu et al. [18] led a study that proposed a novel sequence-to-sequence model based deep-learning framework to recognize complex activities of daily living, leveraging an activity state representation. The proposed activity state representation integrates motion and environmental sensors’ data and reconstructs the motion semantics of the activity.

The SecurHome TV team created a solution only based on the user’s interactions with a TV. This approach seemed to be relevant, since a literature review showed that when the TV has been integrated to monitor the daily activities of the elderly in only two straightforward ways: either it only assumes a display function (to, e.g., show to the user measured levels of blood pressure) or is complemented with additional sensors.

Some projects and solutions that use the TV to display information about the elderly are presented next. The T-Asisto project [19] incorporates teleassistance services with televisions using DTT (digital terrestrial television) broadcasting technology. This Spanish system integrates the set-top box (STB) used in DTT with a tele-assistance terminal that receives alerts from various sensors (gas, smoke, fire, movement, etc.) around the user’s home. The terminal also makes it possible to detect emergency events triggered by the user and to show the user’s text messages (SMS) and appointments on the television. The interactive TV application AAL@MEO [20], funded under the AAL4ALL (Ambient Assisted Living for All) program, allows one to display several pieces of medical information on TV, such as vital sign measurements, weight measurements and informative videos, based on a dashboard with a summary of all the information that is made available to the user [21]. The company BL Healthcare [22] provides a set of telemedicine functionalities through the TV, supported by its own software. Said software can collect data from wireless medical devices (e.g., blood pressure meter and scale), and displays the data in the TV and sends them to medical professionals.

On the other hand, there are several research projects that use the TV as a device in which sensors are installed so that it is possible to monitor the senior’s domestic activity. These sensors are of different types, from electrical current plugs (to know if the senior has turned on the TV), wireless tags (to understand if users have touched the TV or remote control), and infrared sensors to measure temperature or even movement [23,24,25].

An assessment of the potential of digital interactive television (iDTV) to promote original services, formats, and content that may be relevant to support the healthcare and well-being of individuals was also performed during the course of the IDTV Health project [26]. The research team of this project reinforced that several authors suggested that digital interactive television has the potential to deliver health and social care services to people in their homes, based on the systematic review of digital interactive television systems and their applications in the health and social care fields by Blackburn, Brownsell, and Hawley [27]. In this review, the authors addressed a total of 25 projects categorized into seven groups: (i) e-commerce; (ii) health consultations; (iii) health information and education; (iv) infotainment; (v) local services and events; (vi) social interaction; and (vii) monitoring of vital signs. An analysis of this review shows that 68% of the projects described fall into solutions with features related to: monitoring (for example, of vital signs); provision of healthcare and related information; and remote medical consultations. In percentage terms, this high number of projects, with concerns at the level of tele-care, confirms the importance that the scientific community sees in this area. In terms of functionalities, from the set of projects related to tele-care, the communication functions that work between users stand out and were present in all the projects. These are essentially supported by voice calls. In the review, it was clear a trend of systems relying on additional sensors to provide vital signs, monitoring services, and improve, for example, the response times of care networks in emergency cases. Another perceived trend indicated that 30% of the reviewed works included additional devices to facilitate interactions with applications, especially for people with some kind of physical limitation. These interaction mechanisms are especially suited to specific audiences, such as seniors. According to the authors, the problems of the projects were centered on the fact that only 30% of them have been implemented in commercial platforms.

The SecurHome TV ecosystem differs from the solutions identified by Blackburn, Brownsell, and Hawley, as it has the advantage of avoiding the need to install any additional device and is able to detect very specific deviations from the elderly person’s activity pattern: television consumption and interaction with the TV (turning the TV on or off, changing the volume, changing TV channels, reacting to medication reminders, etc., are recorded and considered by the warning algorithm). As the system is supported on commercial IPTV STB, it has also the advantage of smooth integration in the elderly’s households. This is done via a totally remote operation, avoiding the need for any kind of physical installation.

## 3. SecurHome TV Ecossystem

The SecurHome TV ecosystem is supported by a set of three client applications, complemented with an API to access the main data generated by the utilization of its end users (seniors and their caregivers). Regarding the applications, two are used mainly by caregivers (SecurHome Admin and SecurHome Mobile), and the television application (SecurHome TV) is used by seniors. These applications are briefly explained below.

SecurHome TV—installed in senior users’ STB to monitor and record their television interactions and display medical reminders.SecurHome Admin—provided to caregivers, via a Web platform, to manage their dependents and deliver medical reminders.SecurHome Mobile—provided to caregivers, via a Web application, to monitor the home interaction of their dependents and receive alerts.

The API was implemented through an extensive set of access methods, with the purpose of allowing the other partners of the SecurHome Project to obtain the data generated using the SecurHome TV ecosystem applications.

### 3.1. Information Flow and User Scenarios

The SecurHome TV application is aimed at elderly users, and its main objective is to monitor the activity of seniors during their interactions with television content and with the application itself. This application also allows the presentation of reminders related to medication taken by the elderly and the sending of emergency alerts to the caregiver (Figure 1).

It is in the SecurHome Mobile application that the caregivers have access to emergency alerts and the elderly person’s television consumption data (including responses to medication reminders). In this way, caregivers can monitor in real time the activity of the elderly, anticipating possible accidents at home. In this application, it is also possible to send messages to the elderly, which are displayed through the STB where the SecurHome TV application is installed (Figure 1).

From the SecurHome Admin application, the caregiver can manage the profiles of his/her dependents and his/her own profile. In relation to his/her dependents’ profiles, the caregiver can add the senior’s usual medication, and the medical appointments and exams that he/she has scheduled. After scheduling the medication, the respective reminders will be triggered in the SecurHome TV application (Figure 1).

From the description of the information flow between the applications of the SecurHome TV ecosystem, it is possible to identify several user scenarios (described in detail in the validation process) that explain the relevance and usefulness of the applications’ functionalities. Two of these user scenarios are presented below.

User Scenario 1

The caregiver enters his/her dependent’s medication for the week into the SecurHome Admin application (for example: omeprazole at 8:00 a.m. and aspirin at 1:00 p.m.). The senior turns on the TV and their STB at 7:30 a.m. and the SecurHome TV application is resumed. At 8:00 a.m., the senior receives the medication reminder regarding omeprazole. The senior confirms that he/she has already taken the pill. The senior notices that the aspirin has run out and he/she has no pills to take at 1:00 p.m. The senior presses the yellow button on the remote control and sends an emergency warning to his caregiver. The caregiver receives an emergency warning (red) on the SecurHome Mobile application. The caregiver contacts the senior.

User Scenario 2

The senior has a prescription for two medications per day (fluoxetine at 12:00 p.m. and nebivolol at 8:00 p.m.). The senior receives the reminder for the fluoxetine medication at 12:00 p.m. but does not respond to the reminder. About 15 min later, the caregiver receives a yellow alert on the SecurHome Mobile application. The caregiver does not respond to the alert. A new alert is sent with a higher grade (orange), reinforcing the importance of the caregiver contacting his or her dependent to see if everything is okay with him or her.

### 3.2. System Architecture

From a technological point of view, a code-based server-side development scheme was used that is compatible with all applications and allowed the creation of an integrated development environment capable of responding effectively to the respective requirements. The SecurHome TV application was developed with the support of MediaRoom middleware (which is the most widely used IPTV middleware in the world—©MediaKind). To implement the SecurHome TV application, commercial STB from MEO’s commercial IPTV system were used. The fact that these STB are equipped with the Mediaroom middleware implied that the server-side code development was done in C#, and the SecurHome Admin application (backoffice of the whole system) was developed using the. NET framework. The SecurHome Mobile application was also developed using this framework. The SecurHome TV ecosystem was complemented with an SQL database where all the applications’ data are stored and with an API for simplified access to this data.

Figure 2 illustrates the system architecture of the ecosystem, developed in the scope of the SecurHome TV project, where the mentioned components and the relationships between them are represented.

### 3.3. SecurHome TV

The SecurHome TV application consists of a login screen and seven main areas: Login, Health, News, Community, Information, Settings, and Turn Off, which are described below.

#### 3.3.1. Login

Demonstration Video URL: https://tinyurl.com/jhp8brkt (accessed on 12 October 2021)

After loading the application, the user is directed to the access screen and must press OK to enter. Next, the user selects the desired profile and is asked for the access code, which is composed by four digits (Figure 3). It is worth saying that, in the future, this login process should be improved to allow automatic identification of the elderly/elderlies in order to cope with recognition of multiple users [28]. In addition, and despite the already implemented access code scheme and the fact that this system is targeted to be used by elderlies living alone or at the most with their partners (lowering privacy issues), it is the aim of the Securhome TV researchers to find a better solution to cope with the privacy issues identified in [28,29].

If the code is correct, the login operation is completed and the application displays the respective menu (otherwise, the user receives feedback that the PIN is wrong and will have to return to the user’s area of choice).

The central menu provides access to the areas of Health, News, Community, Information, Settings, and Turn Off. Most of these areas are composed of one or two pages, each with several tabs (see Figure 4, with an example for the Health area), and they are activated by buttons with suitable icons (and with the same colors as the areas to which they belong).

#### 3.3.2. Health

Demonstration Video URL: https://tinyurl.com/4cnfvvtn (accessed on 12 October 2021)

The first icon displayed on the Menu corresponds to the Health area. When entering this area, the “my medication” page is automatically selected. Here the user has three tabs to be able to check the medication that is scheduled for the current day and for the next day, and to see the medication that is overdue. In this way, it becomes possible to see the list of medications with details about the times and number of pills to be taken (Figure 5).

The other page in the Health area is the “appointments and pharmacies” page. Here the user has two tabs, one to see which upcoming appointments and exams he has scheduled, and another to view the pharmacies that are closest to his area of residence. A map locating the pharmacy is also displayed. If the button OK is pressed over the map, the user has access to a larger view of where the pharmacy is located and its phone number.

Since this area is mainly intended to be shared with a caregiver, the insertion and editing of information (except for pharmacies whose information is automatically inserted in a geolocation manner) is done through the SecurHome Admin application.

The SecurHome TV application is also associated with medication reminders, which are materialized in notifications that appear on the TV screen. These notifications appear at the time stipulated by the caregiver, in the SecurHome Admin application, for each medication that the respective dependent should take (Figure 6).

Whenever this happens, and there is no interaction between the user and the reminder, the medication automatically moves to an overdue dose, which can be viewed on the corresponding tab (on the “my medication” page).

#### 3.3.3. News

Demonstration Video URL: https://tinyurl.com/h2pddmdn (accessed on 12 October 2021)

The News area is composed of a central page and a shortcut, which are described below.

The central page has three tabs: the “General” tab, where the user has access to the main status information related to himself and his friends (Figure 7); the “News” tab, where messages sent by the caregiver can be found; and the “Requests” tab, where the user has access to the friend requests sent by other users of the application, which he can consult to decide whether to accept or reject. In this area, the user also has access to a shortcut to his scheduled medication.

#### 3.3.4. Community

Demonstration Video URL: https://tinyurl.com/yh8n3t49 (accessed on 12 October 2021)

The features offered by the Community area meet the concept of social networking, but in a simplified way. In this area, it is possible to create and manage a list of friends, mark favorites, and see who is watching television and which programs are being viewed at that moment (Figure 8). In this case, a user can preview the channel his friend is watching and even switch to that channel. At the same time, it is possible to view the personal profile, which includes basic information such as location, age, interests, and skills. It is also possible to make status changes (connected, for those who wish to make the channel they are currently watching available; busy, when the senior wants a greater degree of privacy; and sick). In this area, it is also possible to search and add friends, using a search system, by name, interest, or skill.

#### 3.3.5. Information

Demonstration Video URL: https://tinyurl.com/hbb5rvm9 (accessed on 12 October 2021)

In the Information area, the user has access to utilitarian information, which is sought daily or in situations of need. Thus, in the “meteorology” tab, the user can consult the weather, including civil protection alerts. The senior can view the weather for the same day and for the next three days in his city (Figure 9). By entering one of the days, the user can see in detail the weather information, such as wind speed. In this area, under the “useful contacts” tab, a list of emergency contacts is also displayed, which can be filled in the SecurHome Admin application.

#### 3.3.6. Settings

Demonstration Video URL: https://tinyurl.com/5c3urspn (accessed on 12 October 2021)

On this page, the user has the possibility to adjust the transparency level of the system according to his preferences. This is quite important, considering the overlaps that may arise from the multiplicity of images and actions taking place on the screen. Thus, each user can make the panels of the application have: (i) a completely opaque background, in order to be able to dedicate full attention to the functionalities of the system; (ii) intermediate transparency, where it is possible to see penumbras of the background television programming; and (iii) substantial transparency, for users who do not have difficulty with the simultaneous viewing of a broadcast and the use of the SecurHome TV application (Figure 10).

#### 3.3.7. Turn Off

Demonstration Video URL: https://tinyurl.com/puchnrtc (accessed on 12 October 2021)

From the corresponding button, the user can shut down the application completely (meaning that it will no longer be running in the background) or switch to another user of the household (in which case the application restarts so that the users associated with the household appear on the home screen) (Figure 11).

### 3.4. SecurHome Mobile

The SecurHome Mobile application is launched through login credentials and is composed of five main areas: Dependents, Dependent Details, Television Consumption, Medication, and Messages. These areas are described below and are accompanied by their respective screens.

As shown in Figure 12a, the caregiver needs to log in to the SecurHome Mobile application using their email or username and the password provided by the system. If the user forgets his password, he can always retrieve it via the link provided on this initial screen. In the dependents area Figure 12b, the caregiver can access the dependents’ statuses (on or off) and see if there are any warnings associated with them. To find out more information about a specific dependent, the caregiver needs to click on him/her. In the details area of each dependent Figure 12c, the caregiver can find out more information about the warning associated with the dependent (if the background is green, there are no warnings), and his television consumption and medication data. In this area, it is also possible to access an interface for writing messages to the dependent.

In the television consumption area (Figure 13a), the caregiver can view the last 10 programs watched by the dependent. In the medication area (Figure 13b), the caregiver has the possibility to identify the dependent’s next medication to be taken, and the status of the latest medication (taken or untaken). In the message area (Figure 13c), the caregiver can choose to send a predefined or personalized message to his dependent.

### 3.5. SecurHome Admin

The SecurHome Admin application, like the others, is started via a login system and consists of three main areas: News, Profile, and Health. These areas are described below. Although the SecurHome Admin application is aimed at caregivers, it can also be used by elderly if they have autonomy and digital literacy to access the application.

#### 3.5.1. Login

The caregiver needs to log in to the SecurHome Admin application using their dependent’s e-mail or username and the password provided by the system (Figure 14).

As in the SecurHome Mobile application, if the caregiver forgets his password, he can retrieve it using the link provided on the screen.

#### 3.5.2. News

The news area of the SecurHome Admin application is a replica of the same area of the SecurHome TV application (Figure 15).

#### 3.5.3. Profile

Through the “Your Profile” and “Edit Profile” options (at the Profile area), the caregivers can view and edit their dependents’ profiles (Figure 16).

#### 3.5.4. Health

In the Health area, the caregiver can view the active medication prescriptions that the dependent has, being able to view details regarding the intake of each prescription (Figure 17). The “Medication” option allows the caregiver to associate medications with the dependent’s profile, by mealtime (breakfast (8 a.m.); lunch (12 p.m.); snack (4 p.m.); dinner (8 p.m.); supper (12 a.m.)) or by periodicity (every 30 min with the possibility of adding intervals of 2, 4, 6, 8, 10, and 12 h or once a day). In the “Appointments” and “Exams” sections, the caregiver can schedule these types of actions for his dependent. Finally, in the “History” section, the caregiver can view the dependent’s medication history (medication already taken and untaken). Additionally, in this section, the caregiver has the possibility to mark all the medication presented as taken.

## 4. Warning Algorithm

The way the alert generation algorithm works is sequential and cumulative. All users are analyzed one by one by the system through a set of sequential criteria that, depending on the data analyzed, can simply interrupt the process or make it follow different paths. The SecurHome TV application runs in the IPTV STB that already exists in the elderly’s houses and permanently collects diverse information that allows profiling users activities, such as:Login and logout date in the SecurHome TV application;Viewing history considering TV Channels and TV Shows;Medication History based on reminders triggered on the TV through the SecurHome TV application (response times and missed reminders);History of emergency requests issued through the emergency button on the remote control of the SecurHome TV application (these requests are sent to the caregiver by email and SMS);Medical events calendar (appointments and exams);Interaction history within the application via the TV remote control (records the areas that the user accesses and their respective times of use);Launch and interaction with the presence panel, which, after 40 min without interaction with the TV, asks the user if he is still watching TV, allowing to know, for example, if the user has fallen asleep.

The analysis process of each user goes through three phases. The first phase consists of a general analysis, identical for all users, where some interaction parameters are identified. The second phase considers the user’s status in the TV application (online/offline); and finally, the third phase is the generation and dissemination of the alert. This phasing, which is repeated by a routine every 15 min, is important, since it defines a hierarchy in the criteria, to guarantee code optimization, and consequently, greater speed in user analysis. There are five types of alerts, with different degrees of severity represented by colors. These colors are associated with each dependent’s profile in the SecurHome Mobile application, in order to notify the caregiver of the alert level they are at. A flowchart that illustrates the described process (Figure 18) is presented below.

The different types of alerts are identified and presented to the caregiver as follows:Green: The severity level associated with this color is null, since there is no pattern deviation detected regarding the elderly user activity. For this reason, there is no warning issued to the caregiver.Blue: The severity of the blue alert is low, and it only acts as a signal (which is sent by email to the respective caregiver). This type of alert is issued whenever a minor deviation in the senior’s activity is detected. Generally, this alert is sent to the caregiver when medication is overdue, although other indicators suggest that everything should be fine with the dependent person. This alert is also triggered when there is a small deviation in the user’s response time to the presence panel. When the user has little activity recorded over the last five days on the application but is not watching TV as expected, this type of alert is also sent to the caregiver.Yellow: The degree of severity of this alert (which is sent by email to the respective caregiver) is medium and requires the caregiver’s attention, because the detected deviation must be greater than that determined at the previous alert level (blue). Examples of such instances are a deviation in the response time to the presence panel and when the user has some recorded activity in the last five days but is not watching TV as expected. A yellow alert is also sent if a deviation in the elderly person’s activity pattern falls under alert 2 in addition to overdue medication.Orange: The severity of this alert (which is sent by email and SMS to the respective caregiver) is serious, since the deviation from the standard is large according to analysis levels. In these situations, the caregiver should contact his dependent immediately. Following the algorithm’s analysis logic, the deviation detected is greater than one determined for a yellow alert, due to a longer response time to the presence panel, and when, having a high activity record in the last 5 days, the elderly person is disconnected from the application.Red alert: The severity of this alert (which is sent by email and SMS to the respective caregiver) is the highest, and it is very serious. The caregiver should immediately contact his/her dependent. This warning results mainly from requests for help made by the senior using the emergency button on the remote control.

An example of these deviations is illustrated in Figure 19. On first line (day *n*) the regular daily activity of the elderly is shown, in which the user is watching TV and interacting with the TV and with the application when taking medication.

In the second line (day *n* + 1) of Figure 19, the daily routine is changed, and the algorithm detects a deviation from the elderly person’s usual activity, and alerts are sent to the caregiver through the mobile application. These alerts increase in severity if there is no response from the caregiver.

It is important to note that, if the caregiver does not clear an alert, the degree of that alert increases, and successive alerts are continuously sent to the caregiver until he/she gets a reaction.

The algorithm also considers the schedule of medical appointments (appointments and exams) associated with the dependents, in that the alert level is decreased in situations where users are expected to be away from home.

### 4.1. Validation Process

Before the SecurHome TV ecosystem was made available to real users, and despite the field trial where the core engine was evaluated [7], an extensive set of validation tests of the warning algorithm was carried out. To this end, fifteen scenarios were created to simulate possible behaviors by users (both elderly and caregivers), with regard to the dynamics of the following actions (Figure 20): (i) application activity, namely, the launching of reminders, presence panels, and alerts; (ii) caregiver actions, specifically the responses to the alerts requests and the adding of medical information to the senior’s profile; and (iii) user actions—in particular, the interactions with the application and with the TV (change channel, volume adjustment, mute, play of record TV programs, etc.), the absence of interactions, the responses to the application requests, and the sending of emergency alerts.

In Figure 20, the actions (both from the users as well from the caregivers) are divided by type and day and are numbered in the order they were performed. In the ALERT row, the alerts that were issued by the system are identified by the colors associated with the corresponding levels: (i) green—no warnings; (ii) blue—level 2; (iii) yellow—level 3; (iv) orange—level 4; and (v) red—level 5. When an alert is delivered, it is possible to see which actions triggered the emission of this alert through the corresponding color.

As the algorithm deviations are based on the user’s activity in the last 5 days, the tests were divided by days. On the first day, the caregiver added the usual medication (one pill in the afternoon) and the next medical exams of the dependent (on day 9) so that the application could be functional. It should be noted that the user turned on the TV for about two to three hours in the afternoon, as he was not at home in the morning. In the first 4 days, the user interacted with the TV set and the TV content and responded to all medication reminders and also to the presence panels, not creating any alerts. On days 5 and 6, as the user did not respond to a presence panel and a medication reminder, respectively, the system generated a level 2 alert (blue), since it was possible that he had just fallen asleep or left the TV on; however, the reason could have been more serious, so an alert was sent to the caregiver. In these cases, the caregiver viewed the alerts and verified that everything was fine with the dependent and cleared the alerts. On day 7, as the user did not respond to a medication reminder and to a presence panel, cumulatively, the system generated a level 3 alert (yellow), since a more serious situation than those described above could have occurred. On day 8, the user generated an emergency alert (level 5—red) through the application (pressing the yellow button on the remote), and the caregiver responded to this senior’s help request. On day 9, the user had medical exams scheduled for the afternoon. Since he was not at home, he did not turn on the TV, but the system launched the medication reminder. The system did not generate any alert for the caregiver, as it realized that the senior was not at home (medical exams were scheduled in the application). On the 10th day, the user still did not turn on the TV in the afternoon, as he went to an unexpected medical appointment. However, as the caregiver did not update the application with this information, the system continued to generate alerts. On that day, the system already had overdue medication reminders, so when it got another unanswered reminder, it launched a level 4 alert (orange). As the caregiver went with the senior to the medical appointment, he did not respond to the alert and the system launched another alert, this time a level 5 alert (red). The caregiver ended up clearing the alert. On day 11, no alerts were generated, because the user actively interacted with the TV and with the application itself (no presence panel was launched) and responded to the medication reminder that was launched by the system. On day 12, the user did not respond to a presence panel and the system generated a level 2 alert, as it had done on day 5. On day 13, the user did not respond to two system requests: a medication reminder and a presence panel, and for that reason, the system generated a level 3 alert, like what happened on the 7th day. On day 14, the user generated an emergency alert through the application, and as the caregiver did not respond to this request for help from the user, the system alerted the caregiver with another red alert. Finally, on the 15th day, no alerts were generated, since the user responded to all application requests and maintained continuous interaction with the television and the application.

### 4.2. SecurHome API TV

The warning algorithm, described above, is based on a server application developed with the purpose of analyzing the data collected by the TV application and detecting deviations from the stated criteria. Based on the severity of these deviations, alerts are generated and sent to caregivers with the intention of alerting them to emergency situations with the elderly.

In its current version, the algorithm has some limitations regarding constraints while setting the user’s patterns—namely, upon cold start of using the application and when the user’s routines change considerably when compared to the pattern established in the previous 5 days. These limitations may impact the levels of alerts issued, decreasing the effectiveness of the system. In the Discussion, these questions are pointed out.

In this type of scenario, it would be very relevant to combine more criteria and process them with machine learning logic. To this end, an application programming interface (API) was implemented with an extensive set of access methods, with the aim of allowing the other partners of the SecurHome project to obtain the criteria presented in Table 1 and Table 2. The API was implemented in collaboration with the University Carlos III of Madrid (UC3M) partner, so that the data obtained could be processed in the respective infrastructure that benefits from a network of sensors.

The Web API is accessed through the following URL: https://meoapp.ptinovacao.pt/2020/Prod/securhomeua/webapi/api/ (accessed on 12 October 2021), which returns a JSON file (see Appendix A to check some examples).

Regarding the data associated with the use of the SecurHome TV application, the API gives access to the senior’s personal, interaction, and medical data (Table 1).

In relation to the data associated with the use of the SecurHome Mobile application, the API gives access to the caregiver’s personal data and their access data (Table 2).

### 4.3. API Usage by the Project Partner

A new API was developed by the UC3M partner considering the structure of the SecurHome TV API to support a sensory device for homes (SDH) [30] based on an IoT system, which collects information non-invasively through a network of sensors. It uses a set of sensors that allows it to collect information about the user’s environment and physical behavior. This data is stored, and the IoT system uses the data to detect anomalies in the user’s behavior and notify his caregivers. In addition, with the use of this API, the device is able to know the user’s medication and medical appointment routines, being able to generate notifications, either to remind them to take medication or that they have a medical appointment. This last reminder is done twice, the first time the day before the appointment, and the second time two hours before the appointment. It is important to mention that the time of each reminder can be modified. Some examples of these notifications are shown in Figure 21.

It is important to mention that the SecurHome TV API was used to structure the database of the new API of the UC3M partner, considering the data that the first one provided, with the objective that the data storage structures in both systems should be compatible, or at least very similar, simplifying future communication between systems and reinforcing the warning algorithm of SecurHome TV.

The API has been extended so that the system, in addition to considering the medication routine, medical appointments, and all the information provided by the SecurHome TV ecosystem for sending alerts, also considers the environmental information of the user’s home to generate alert messages about anomalies in the person’s physical behavior or environment. The same user–caregiver structure proposed in this paper has been used (including detailed information about both types of users) and the alerts detected by the device are reported via e-mail to the user’s caregivers.

The following is a brief description of the different services presented by the extended API.

Storage of user data—services to store:User data in the database, user status, and user login information;Information of TV programming watched by the users;Messages sent by users’ caregivers and received by users;Alerts detected by the University of Aveiro in its users;Medical appointments and exams of the users;Medication that users have to take;Types of appointments and exams.
User data—services that display:A list of all users;The personal data of a specific user;A list with the caregiver-dependent dependency;The medication to be taken by a specific user;The medical appointments that a specific user has;The types of medical appointments that are registered;The status of the users;The login data of the users;The TV programming watched by the users;Messages sent by users;Messages received by users;Users’ alerts.
Sensor data—services to:Store data collected by sensors in the database;Display a list of sensors used by the home IoT device;Display the data collected by the sensor network from a specific id;Display the data collected by a sensor and a specific user in a defined period of time.
Rules and alerts—services to:Store a new alert rule;Display a list of alert rules and relationships between rules for detecting unusual behavior of a specific user;Store alerts triggered by the detection of unusual user behavior.

## 5. Discussion

In this section, both strengths and weaknesses of the SecurHome TV ecosystem are addressed in order to upgrade it in a future release.

With the validation, it was possible to verify that the algorithm is able to detect most of the foreseen situations that can indicate a potential problem with the senior. However, as the algorithm is based on the senior’s activity, historical data of a minimum of 5 days is needed as a basis for analysis. Additionally, if a medical appointment or exam is not registered by the caregiver in the system, the algorithm can create a false alarm if in that period of time it is normal for the elderly person to have registered activity, since it cannot understand that the elderly person is not at home. Likewise, if the senior suddenly changes their routine (e.g., changing TV viewing periods), the algorithm may initially detect false alerts, since a deviation from the daily activity pattern is momentarily created.

Additionally, efforts were made by the UC3M project partner to extend the developed API, granting the connection between a dedicated IoT module [30] and the SecurHome TV ecosystem, enhancing the former with additional information on the user’s behavior and home environment. This same work was also of great importance for the conceptualization of the IoT module, because the initial API allowed the execution of pilot tests regarding the generation of notifications for medical appointments and medication to the user.

As stated earlier, it is possible to query the user’s data of the SecurHome TV ecosystem in order to achieve integration of data from both systems, allowing that in the future, elderly users will be able to decide to have both systems in their homes, and thus hazard situations could be detected more reliably.

In addition, the privacy problems and the automatic user recognition strategy are issues that should not be neglected in a future upgrade.

## 6. Conclusions

This paper described in detail the SecurHome TV ecosystem, the novelties of its approach when compared with the solutions addressed in the literature review, and the validation process of its warning algorithm.

The SecurHome TV ecosystem allows, without additional sensors and in a transparent way, detecting potential hazard situations and alerting the senior’s caregivers to take immediate action. This dynamic is based by monitoring the senior’s domestic activity through the TV set and detecting deviations from his/her interactions with and TV consumption pattern through a dedicated warning algorithm. The results of the validation showed that, not only can the algorithm detect most potentially risky situations for the elderly, but also that this non-invasive approach has a role to play in keeping an eye on elderly well-being at home.

Additionally, the integration of the SecurHome TV algorithm and related data with the IoT module developed by UC3M project partner has the potential to detect hazard situations in a more reliable way. In this sense, the reported (slight) limitations may be overcome with future work benefiting from the effective integration and evaluation of the two systems (SecuHome TV + IoT module).

## Figures and Tables

**Figure 1 sensors-21-06897-f001:**
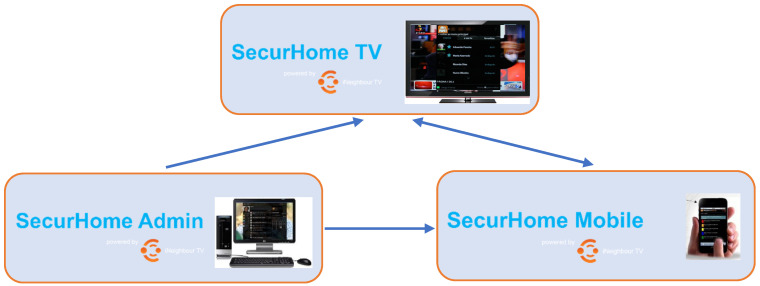
Information flow of the SecurHome TV ecosystem.

**Figure 2 sensors-21-06897-f002:**
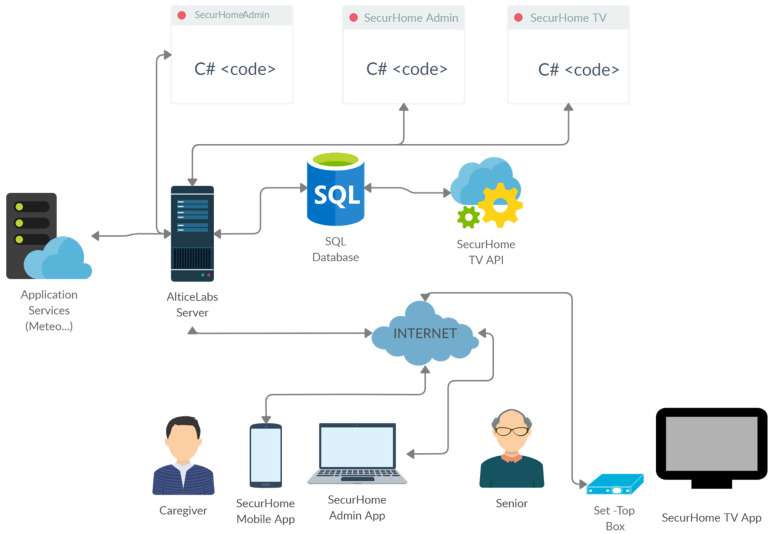
System architecture of the SecurHome TV ecosystem.

**Figure 3 sensors-21-06897-f003:**
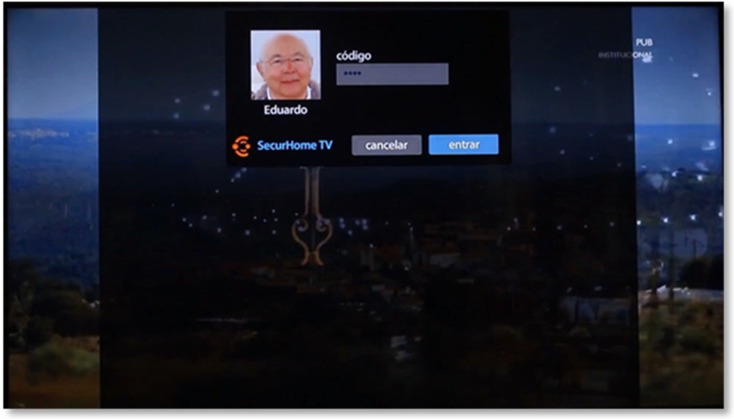
SecurHome TV application login.

**Figure 4 sensors-21-06897-f004:**
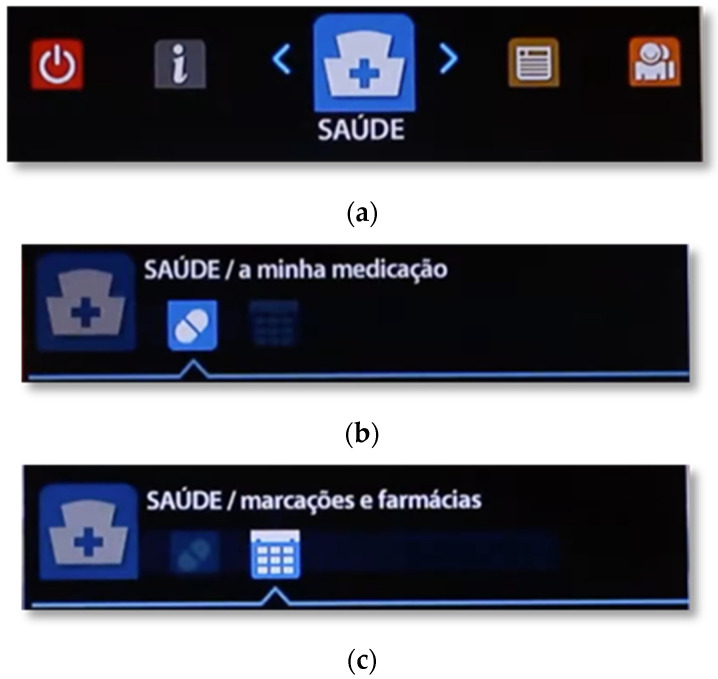
Breakdown of the main menu into areas and their tabs: (**a**) Menu—Health area. (**b**) Health area—“my medication” page. (**c**) Health area—“appointments and pharmacies” page.

**Figure 5 sensors-21-06897-f005:**
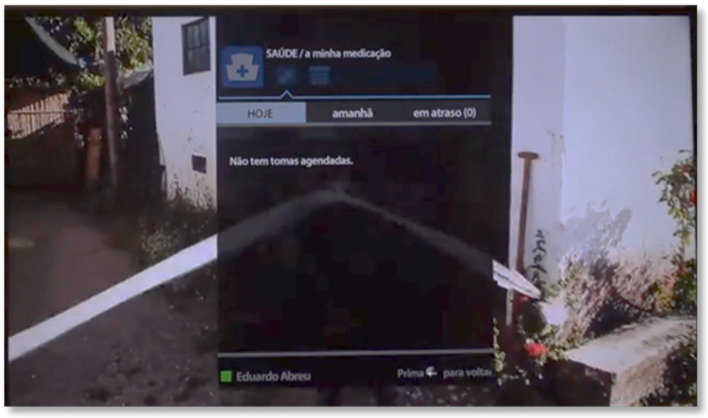
Medication scheduled for today in SecurHome TV application.

**Figure 6 sensors-21-06897-f006:**
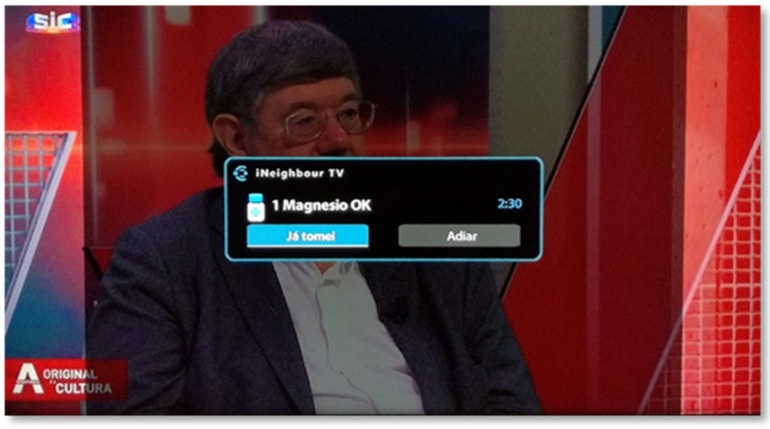
Reminder to take medication.

**Figure 7 sensors-21-06897-f007:**
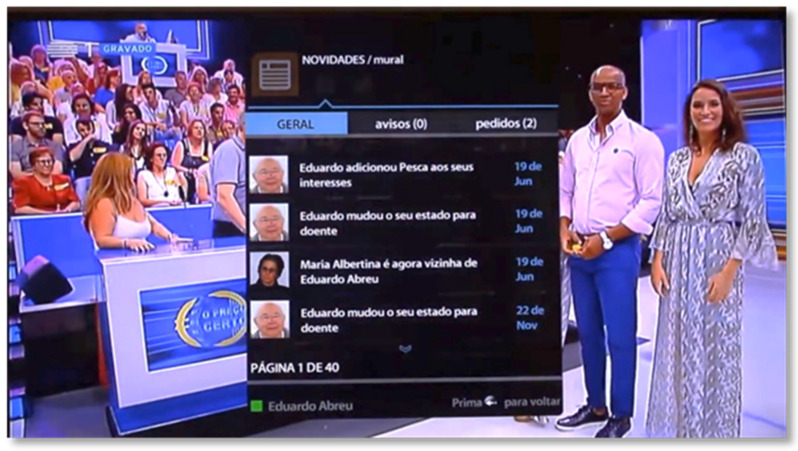
News area of the SecurHome TV application.

**Figure 8 sensors-21-06897-f008:**
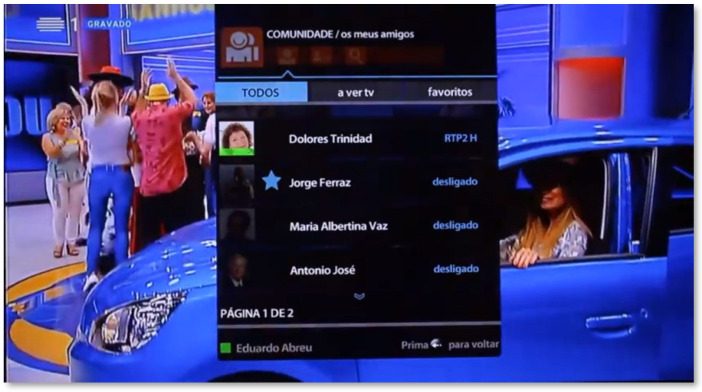
Community area of the SecurHome TV application.

**Figure 9 sensors-21-06897-f009:**
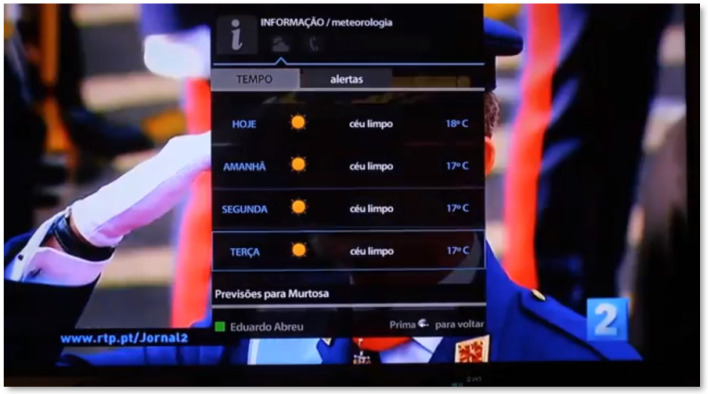
Information area of the SecurHome TV application.

**Figure 10 sensors-21-06897-f010:**
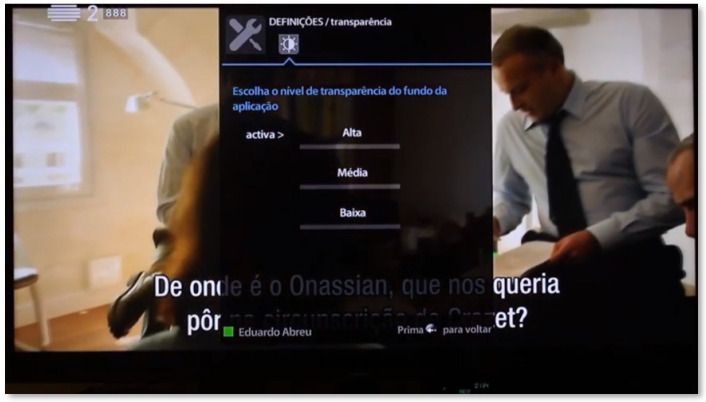
Settings area of the SecurHome TV application.

**Figure 11 sensors-21-06897-f011:**
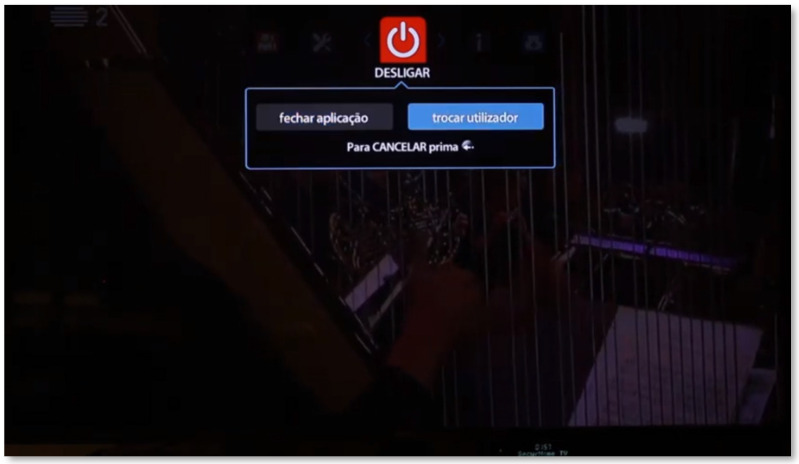
Turning off SecurHome TV.

**Figure 12 sensors-21-06897-f012:**
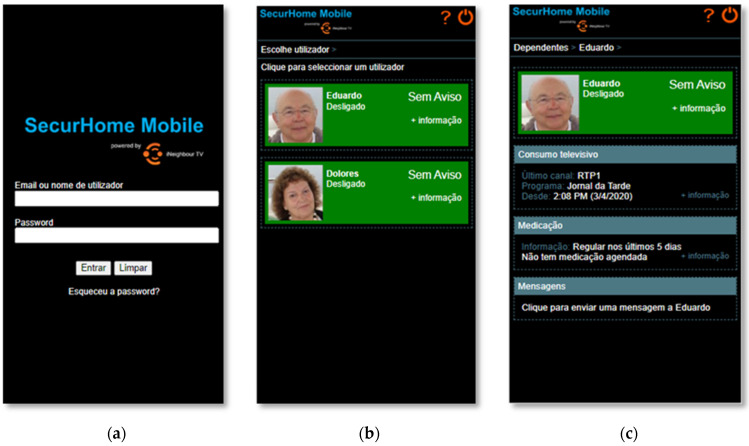
(**a**) Home screen of SecurHome Mobile; (**b**) the Dependents area of SecurHome Mobile; (**c**) the Dependent Details area of SecurHome Mobile.

**Figure 13 sensors-21-06897-f013:**
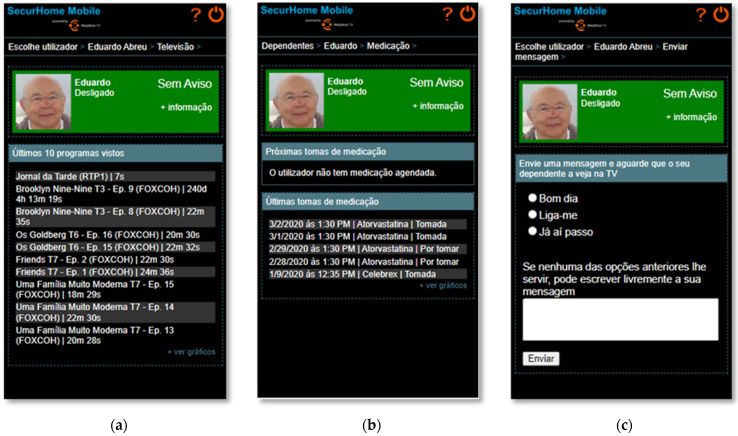
(**a**) Television consumption of a dependent viewed in the SecurHome Mobile application. (**b**) The medication schedule of a dependent in SecurHome Mobile. (**c**) The message sending area of SecurHome Mobile.

**Figure 14 sensors-21-06897-f014:**
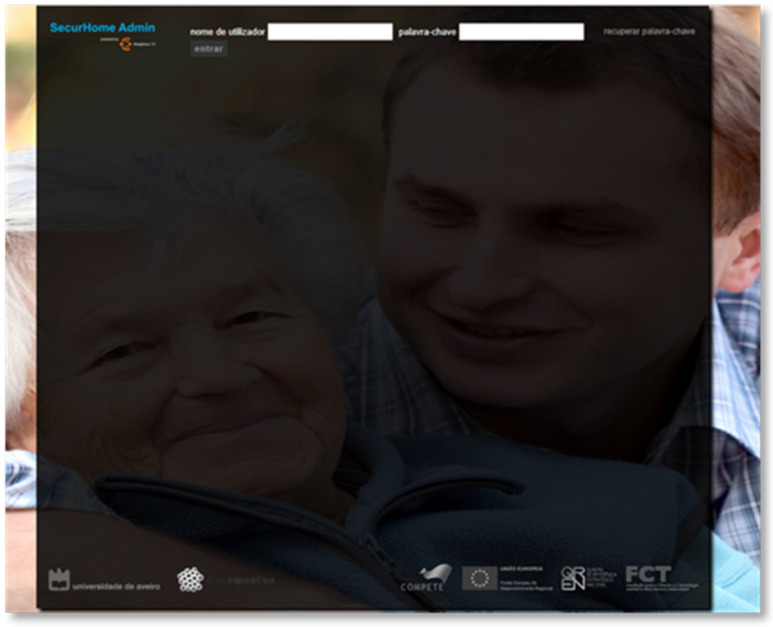
Login screen of the SecurHome Admin application.

**Figure 15 sensors-21-06897-f015:**
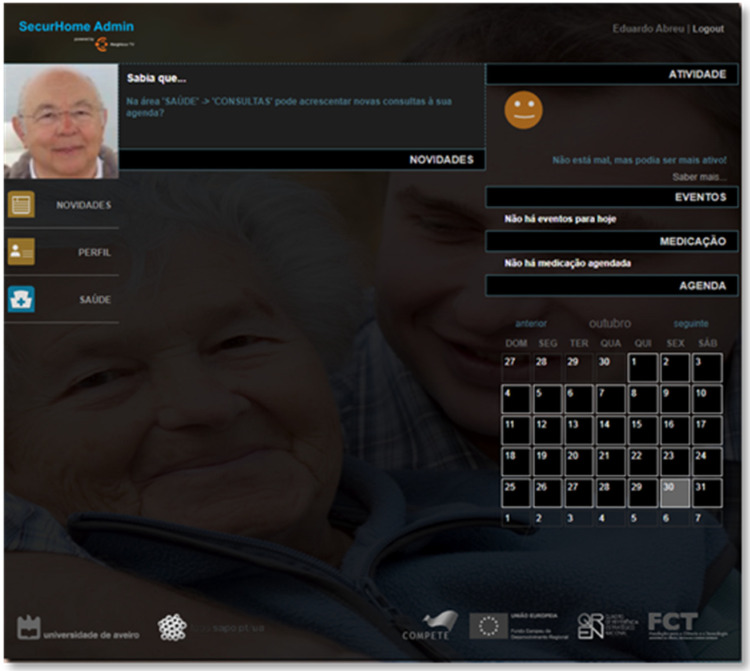
News area of SecurHome Admin.

**Figure 16 sensors-21-06897-f016:**
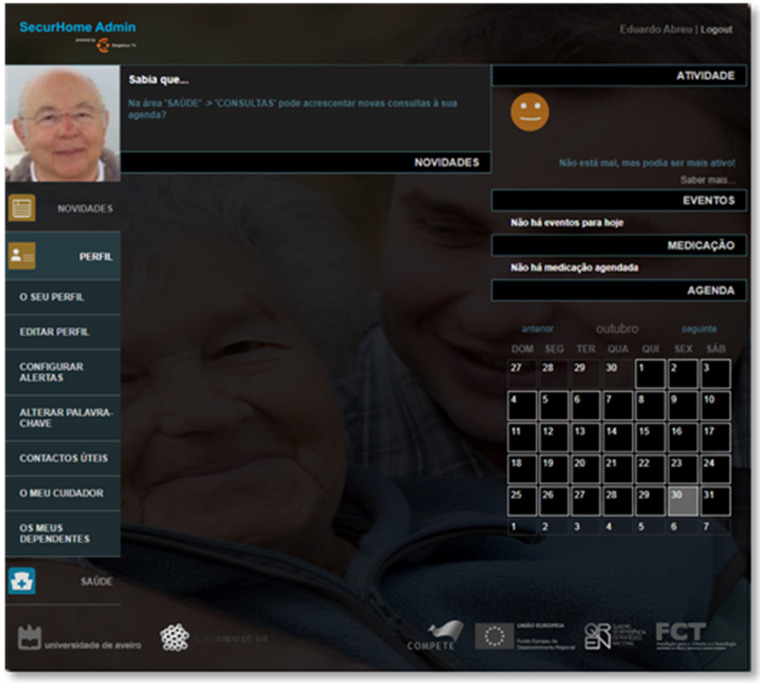
Profile area of the SecurHome Admin application.

**Figure 17 sensors-21-06897-f017:**
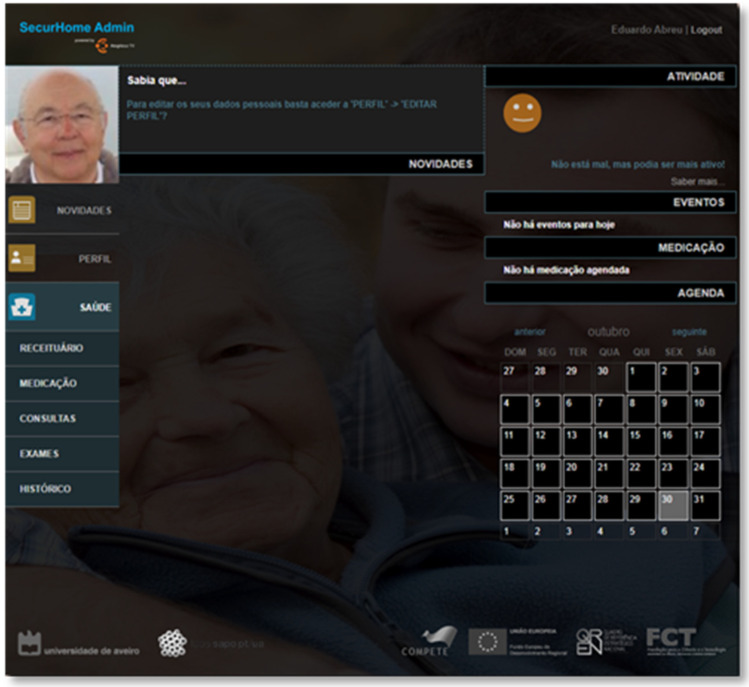
Health area of SecurHome Admin application.

**Figure 18 sensors-21-06897-f018:**
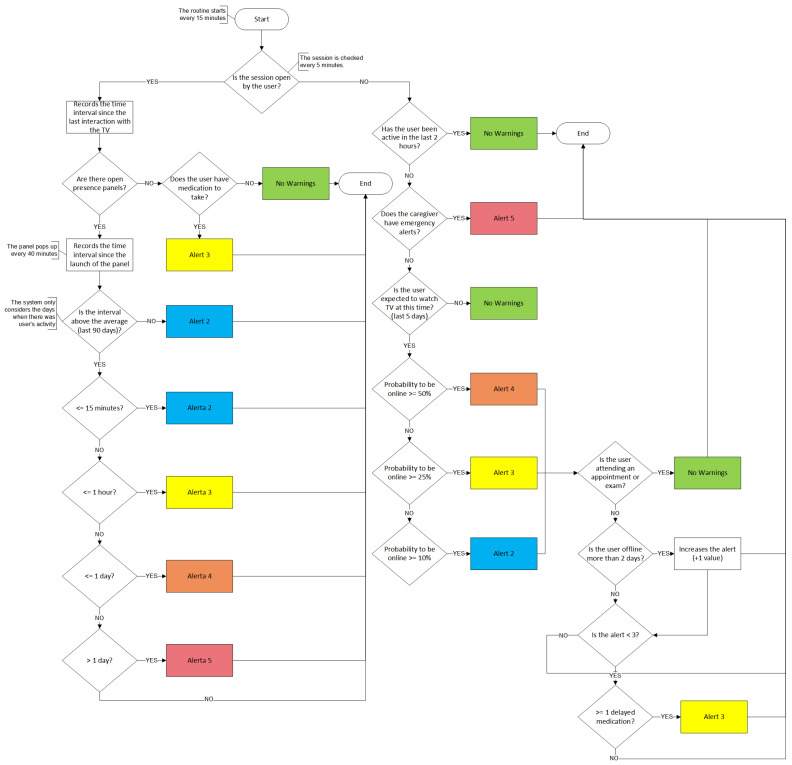
Analysis process of the warning algorithm.

**Figure 19 sensors-21-06897-f019:**
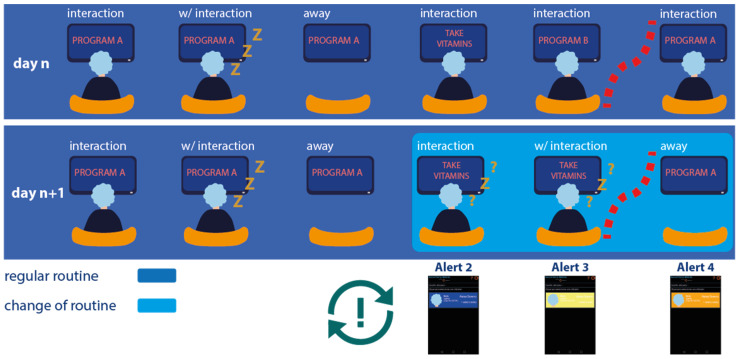
An example of a deviation in an elderly person’s domestic activity.

**Figure 20 sensors-21-06897-f020:**
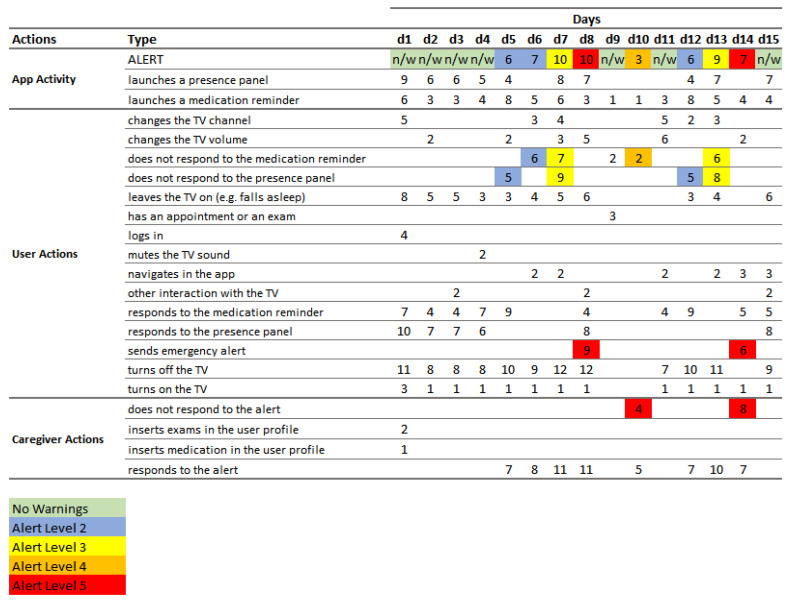
Analysis grid of the warning algorithm’s behavior.

**Figure 21 sensors-21-06897-f021:**
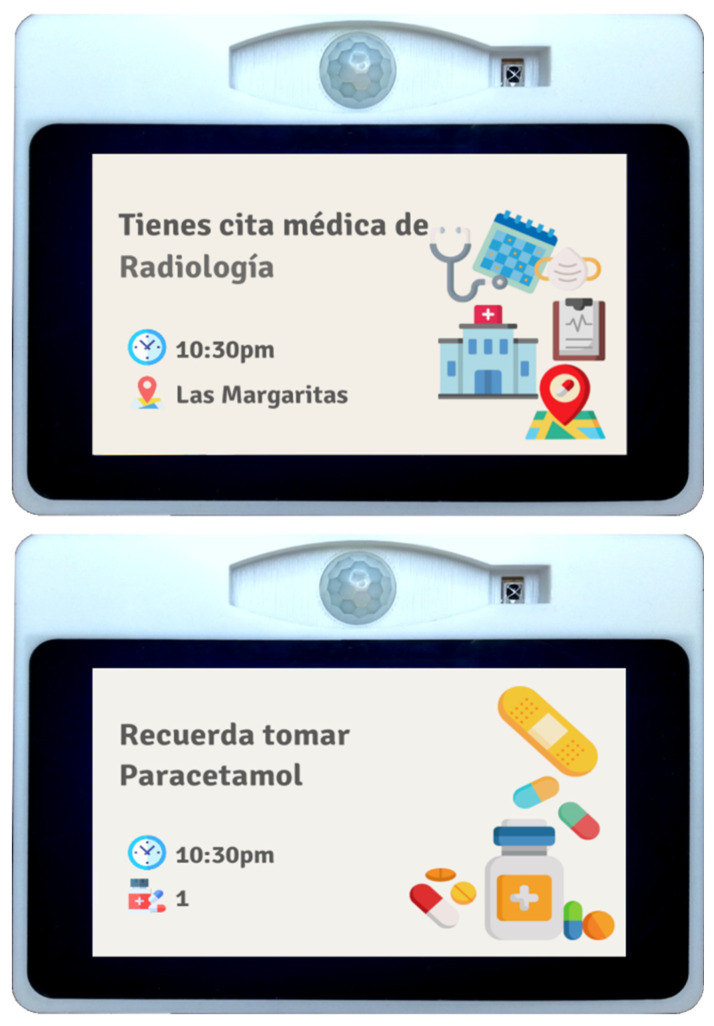
Home IoT device notification screen.

**Table 1 sensors-21-06897-t001:** Data Associated with the use of SecurHome TV application.

Category	Type	Data
Personal Data	User Profile	name, sex, birth date, city, profile photo, hobbies e skills
	Current Warnings/Alerts	
	Received Messages	
Interaction Data	API Status	offline, online
	Time of the Last Interaction with the TV	
	Real-time Television Consumption	TV channels and programs being watched
	Television History	last programs viewed
Medical Data	List of Medications and Dosage	
	List of Upcoming Medical Appointments	
	List of Scheduled Exams	
	Medication History	time when medication was taken and status (taken/not taken)

**Table 2 sensors-21-06897-t002:** Data associated with the use of the SecurHome Mobile application.

Category	Type	Data
Personal Data	User Profile	name, sex, birth date, city, profile photo
	Dependent List	
	Sent Messages	
Access Data	Login List on Web/Mobile Applications	
	Time of Last Login on the Web/Mobile Applications	

## Data Availability

The data presented in this study are available in Section 4.1. Validation Process.

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
