# Peer review of "TV Interaction as a Non-Invasive Sensor for Monitoring Elderly Well-Being at Home"

_sensors, 2021, doi:10.3390/s21206897_

Round 1

Reviewer 1 Report

I found the article in general interesting and of high quality. It present a functional platform, with examples that convey good performance. In general my opinion would be to accept it with the proposal of some minor changes.

Author Response

Thank you very much for the review. The authors made considerable changes in the manuscript. The authors will appreciate a new review of this version. 

Reviewer 2 Report

This paper describes in detail the SecurHome TV ecosystem. Overall, this paper is timely and the topic is interesting. However, the reviewer has the following comments to improve the quality of this paper:

  1. The authors should add more descriptions in the fouth paragraph in Section I to reveal the novelty of this paper, as compared to existing works.
  2. The literature review should be classified for improving the readability of Section II.
  3. The text in some figures is too small, such as Fig. 2, the authors are suggested to adjust the size of text in figures.
  4. Some case study should be included in this paper for evaluate the performance of SecurHome TV ecosystem.
  5. Some recent works should be cited in this paper, such as

[1] Energy-efficient cooperative communication and computation for wireless powered mobile-edge Computing[J]. IEEE Systems Journal, 2020: 1-12, doi: 10.1109/JSYST.2020.3020474.

Author Response

The authors should add more descriptions in the fouth paragraph in Section I to reveal the novelty of this paper, as compared to existing works.

The authors inserted a paragraph that reveal more clearly the novelty of the SecurHomeTV (lines 63-73).

The literature review should be classified for improving the readability of Section II.

This advice is very relevant, however Sensors guidelines do not foresee it. The authors follow a narrative literature review and made an objective analysis of the topic (remote monitoring and tele-care systems for elderly) and positioned SecurHome TV within this review.

The text in some figures is too small, such as Fig. 2, the authors are suggested to adjust the size of text in figures

The authors adjusted the text of Figure 2, however all figures have an acceptable resolution that allows reading the text with zoom in.

Some case study should be included in this paper for evaluate the performance of SecurHome TV ecosystem.

This suggestion is very pertinent, and the authors clarify the fact that the core engine of the SecurHome TV was already evaluated in a field trial. So, the authors included a phrase with a reference (line 552-553).

Some recent works should be cited in this paper, such as

[1] Energy-efficient cooperative communication and computation for wireless powered mobile-edge Computing[J]. IEEE Systems Journal, 2020: 1-12, doi: 10.1109/JSYST.2020.3020474.
The authors read this very interesting paper in detail, but they were not able to reflect it in the scope of the SecureHome TV article.

Reviewer 3 Report

the authors present the SecurHome TV ecosystem, a technical solution based on the elderly interaction with the TV set - ne of the most common devices in their daily life - acting as a non-invasive sensor enabling to detect potential hazard situations through an elaborated warning algorithm. However, some point needs to be improve 

1- page 6, authors try to write scanarios as points and this give look as report not artice. so i suggest authors to make it as flowchart or explain it as pragraphs 

2- authors are missing most important things that is activity recognization and i suggest authors to read "Multi-user activity recognition: Challenges and opportunities' it is more related and must be discussed in article 

4- conclusion should be one pragraph only to explain the outcome of this study. 

5-authors need reorgnized figures and tables accoridng required tamplete . some figure out of the mergine artice 

6- i am highly recommand this article to be publish after considering my abovepoints in order to make it more readable 

Author Response

page 6, authors try to write scanarios as points and this give look as report not artice. so i suggest authors to make it as flowchart or explain it as pragraphs 

The authors explained the scenarios as paragraphs and added a note that sends the reader to the validation process section, in which more scenarios are described (lines 246-247).

authors are missing most important things that is activity recognization and i suggest authors to read "Multi-user activity recognition: Challenges and opportunities' it is more related and must be discussed in article

The authors considered this article and added it as reference. A paragraph about activity recognition were added (294-300).  

conclusion should be one pragraph only to explain the outcome of this study.

We tried to take this advice in consideration, however it collides with other reviewers’ suggestion (it asked the authors to add more details in the discussion and conclusion sections).

authors need reorgnized figures and tables accoridng required tamplete . some figure out of the mergine artice

The authors followed the template and verified other published articles in the journal and the figures can exceed the text margins when they are large in width.

Reviewer 4 Report

The authors raise the key issues from the point of view of the quality of life. The problem of monitoring the elderly, especially in the face of an aging society, is very timely and very future-proof. It is worth noting that the authors present a solution based on a TV set. This means that there is no need to install special sensors or additional devices. You only need to install their application.
The authors present the state of art, accurately using literature sources.
Most of the article presents the functionality of the developed application. They also include sample scenarios and screenshots. From this point of view, I have doubts whether it is a research work or rather an engineering one, which resulted in a good or very good application. Moreover, the presentation of the algorithm referred to by the authors is not very clear. I think it could be more attractive.
In addition, the question arises as to how the creators thought and implemented the mechanisms of protection against fraudulent login attempts by unauthorized persons, as well as the security of data integration. This is especially important, e.g. when it comes to taking medications, etc.
In my opinion, the discussion presented by the authors is minimal and does not really add much to the article. It is able to be there because it has to be in the article.

Authors must decide whether to SecureHome or SecurHome.

In sum. Compiled by the authors is interesting, but according to my comments, I'm not sure if it is suitable for publication in this Journal. If the editor says yes, then it is certainly necessary to extend the discussion and answer security questions. It would also be good for the algorithm itself, which is de facto the only scientific value in this article, to be better presented.

Author Response

Moreover, the presentation of the algorithm referred to by the authors is not very clear. I think it could be more attractive.

In section 4, the authors added a flowchart that presents the analysis process of the warning algorithm and a figure that illustrates an example of a deviation of an elderly’s domestic activity.

In addition, the question arises as to how the creators thought and implemented the mechanisms of  protection against fraudulent login attempts by unauthorized persons, as well as the security of data  integration. This is especially important, e.g. when it comes to taking medications, etc.

This comment is very important, and the reviewer is absolutely right. The authors clarified the login process and the related security questions (lines 294-300). However, it is worth to say that the use of the system is essentially for a single person or a couple, that is why there is no major privacy problems.

In my opinion, the discussion presented by the authors is minimal and does not really add much to the article. It is able to be there because it has to be in the article.
The authors improve the discussion section with more text, making an introduction and a conclusion.

Authors must decide whether to SecureHome or SecurHome.

In fact, the correct name is SecurHome, the authors revised the seven incorrect entries.

Round 2

Reviewer 4 Report

Paper is able to be published.